# Recombinant Mitochondrial Genomes Reveal Recent Interspecific Hybridization between Invasive Salangid Fishes

**DOI:** 10.3390/life12050661

**Published:** 2022-04-29

**Authors:** Evgeniy S. Balakirev

**Affiliations:** A.V. Zhirmunsky National Scientific Center of Marine Biology, Far Eastern Branch, Russian Academy of Sciences, Vladivostok 690041, Russia; esbalakirev@mail.ru; Tel.: +7-423-231-0905

**Keywords:** clearhead icefish *Protosalanx chinensis*, short-snout icefish *Neosalanx tangkahkeii*, Salangidae, mitochondrial genome, recombination, invasive species hybridization

## Abstract

The interspecific recombination of the mitochondrial (mt) genome, if not an experimental artifact, may result from interbreeding of species with broken reproductive barriers, which, in turn, is a frequent consequence of human activities including species translocations, habitat modifications, and climate change. This issue, however, has not been addressed for *Protosalanx chinensis* and other commercially important and, simultaneously, invasive salangid fishes that were the product of successful aquaculture in China. To assess the probability of interspecific hybridization, we analyzed the patterns of diversity and recombination in the complete mitochondrial (mt) genomes of these fishes using the GenBank resources. A sliding window analysis revealed a non-uniform distribution of the intraspecific differences in *P. chinensis* with four highly pronounced peaks of divergence centered at the *COI*, *ND4L*-*ND4*, and *ND5* genes, and also at the control region. The corresponding divergent regions in *P. chinensis* show a high sequence similarity (99–100%) to the related salangid fishes, *Neosalanx tangkahkeii* and *N. anderssoni*. This observation suggests that the divergent regions of *P. chinensis* may represent a recombinant mitochondrial DNA (mtDNA) containing mt genome fragments belonging to different salangid species. Indeed, four, highly significant (pairwise homoplasy index test, *P* < 0.00001) signals of recombination have been revealed at coordinates closely corresponding to the divergent regions. The recombinant fragments are, however, not fixed, and different mt genomes of *P. chinensis* are mosaic, containing different numbers of recombinant events. These facts, along with the high similarity or full identity of the recombinant fragments between the donor and the recipient sequences, indicate a recent interspecific hybridization between *P. chinensis* and two *Neosalanx* species. Alternative hypotheses, including taxonomical misidentifications, sequence misalignments, DNA contamination, and/or artificial PCR recombinants, are not supported by the data. The recombinant fragments revealed in our study represent diagnostic genetic markers for the identification and distinguishing of hybrids, which can be used to control the invasive dynamics of hybrid salangid fishes.

## 1. Introduction

### 1.1. Taxonomy and Distribution

The family Salangidae Bleeker 1859 comprises around five to seven genera (*Salanx* Cuvier 1816, *Leucosoma* Gray 1831, *Salangichthys* Bleeker 1860, *Hemisalanx* Regan 1908, *Protosalanx* Regan 1908, *Neosalanx* Wakiya and Takahashi 1937, and *Neosalangichthys* Fu, Li, Xia, Lei 2012) and approximately 20 species listed in the Eschmeyer’s Catalog of Fishes [1], the World Register of Marine Species [2], and the FishBase [3]. These fishes (referred to as icefish or noodlefish) are endemic to East Asia and inhabit a wide range of marine, brackish-water, and freshwater habitats in China, Vietnam, Korean Peninsula, Japan, and Russia [4,5,6,7,8,9,10,11]. Salangids are small (reaching a maximum body length of about 21.0 cm), neotenic (adults retain some larval traits) fishes with early maturation, relatively high fecundity, and a life span of about one year [8,12,13]. Although salangids have been studied for more than 100 years (see the references above), the classification and phylogenetic relationships within this family remain poorly resolved, which could be explained in part by their neoteny, high morphological plasticity, and the homoplasious morphological characters used in previous studies [14]. The taxonomic nomenclature of salangid fishes, based on morphological characters and genetic approaches, has been subject to various changes with multiple known synonyms [15,16,17,18,19,20]. Thus, among other taxonomic reorganizations, Zhang et al. [19] showed that *Protosalanx hyalocranius* (Abbott, 1901) is a synonym of *P. chinensis* (Basilewsky, 1855), while *Neosalanx taihuensis* (Chen, 1956) is a synonym of *N. tangkahkeii* (Wu, 1931). Guo et al. [20] found *Hemisalanx* to be a junior synonym of *Salanx*. Fu et al. [18] questioned the generic status of *Neosalanx* and argued that it should be considered as a junior synonym of *Protosalanx*.

### 1.2. Genetics

The currently available genetic data were obtained using the mitochondrial and nuclear DNA markers [21,22,23,24,25,26,27,28,29]. Complete mt genomes were sequenced for *P. hyalocranius* [30], *N. anderssoni* [31], *N. tangkahkeii* [32], and other species (see Appendix A for the accession numbers and a full citation list). The complete nuclear genome was sequenced for *P. chinensis* [33,34,35]. Genetic diversity, population structure, and demographic history of salangid fishes were considered in a series of population genetic studies [28,36,37,38,39,40,41,42]. The problem of fish mislabeling was addressed using the *CytB* gene [43].

### 1.3. Species Transplantation and Introduction

Wild populations of salangids have markedly declined in recent years due to overexploitation, hydrotechnical constructions, and water pollution [44,45,46,47], which stimulated the development of fish aquaculture (review by Kang et al. [48]). Among salangids, the clearhead icefish *P. chinensis* and the short-snout icefish *N. tangkahkeii* are commercially important aquaculture species. In China, they have been successfully transplanted from Lake Taihu (eastern China) to hundreds of lakes and reservoirs across the country [48,49,50,51]. Special artificial breeding techniques have been developed, with fry released into lakes and reservoirs to improve aquaculture yield [52]. The introduction of icefishes has brought significant economic benefits but, simultaneously, jeopardized biodiversity in native assemblages [48,53,54]. Both *P. chinensis* and *N. tangkahkeii* are reported to be successful invasive species [47,48,53,55,56,57].

In Russia, *P. chinensis* was first recorded from Lake Khanka in 2006 and from the Amur River in 2008 [58], which came as a result of its introduction from Chinese water bodies [13,59,60,61,62]. *P. chinensis* is considered a potentially aggressive invasive species preying mainly on larvae or juveniles of indigenous fishes [63]. The same has been observed and raised serious concerns in China [47,48,64]. Indeed, *P. chinensis* is a successful invader: in around three decades [48] the species has substantially expanded its range from Lake Taihu northward up to the coastal waters of the Sea of Okhotsk (the Amur River estuary, Russia) (https://todaykhv.ru/news/in-areas-of-the-province/30507/ (accessed on 8 April 2022)). The distance between Lake Taihu and the Amur River estuary is almost 3000 km.

Transplantations and introductions of non-endemic species frequently cause interspecific hybridization and genetic introgression between introduced and native species with significant ecological and evolutionary consequences for native populations (for review, see [65,66,67,68,69]). A less frequent phenomenon is the interspecific hybridization between invasive species in non-native areas [70,71,72,73,74,75,76,77] with limited or no hybridization in their native ranges. In this case, the hybrid between the two invasive species shows a potential to drive the emergence of new genetic diversity, thus, reducing the effect of inbreeding depression and increasing the opportunities to rapidly adapt to new environmental conditions, which further contributes to invasion success (e.g., [77,78]).

Interspecific hybridizations create the opportunity of mtDNA paternal leakage because interspecific genetic differences (involving both mitochondrial and nuclear DNA) can be large enough to escape the mechanisms destructing paternal mitochondria and leading to heteroplasmy [79,80,81], with possible subsequent recombination (review by, e.g., [82,83,84,85]). Interspecific mtDNA recombination has been reported for a broad range of organisms including hybridizing yeasts [86], brown algae [87], conifers [88], reef building corals [89], salmonid [90,91,92] and cyprinid [93,94] fishes, and primates [95]. Thus, an analysis of mtDNA recombination can be informative for detecting interspecific hybridization, especially for those species which, as with salangids (Section 1.1), are poorly distinguished based on morphological criteria.

As a result of the intentional introduction, the possible hybridization of icefishes (along with some other negative environmental impacts) was suggested [48,53], though not verified by genetic approaches so far. The concern is not ungrounded, because, even without a gene flow, invasive hybridization will likely cause detrimental consequences for the interbreeding species [67,96]. Using sufficiently long mt genome fragments (8141 bp), we previously described a detailed architecture of recombinant events due to anthropogenic hybridization between salmonid fishes such as Siberian taimen *Hucho taimen* and two lenok subspecies, *Brachymystax lenok* and *B. lenok tsinlingensis* [92]. In the present work, we analyzed the patterns of nucleotide diversity in complete mt genomes of icefishes, *P. chinensis* and *N. tangkahkeii*, as well as other closely related salangid species, and detected clear signals of mtDNA recombination. We argue that the revealed mt recombinants are not experimental artifacts but reflect the interspecific hybridization between the icefishes studied. The data suggest that even a relatively high genetic divergence (up to 8.5% between *N. tangkahkeii* and *P. chinensis*) could not have provided any successful reproductive isolation between the species. The results can be useful for the development of responsible aquaculture and biodiversity conservation management practices to minimize the probability of anthropogenic hybridization that can facilitate the invasion process and threaten the resilience of native populations.

## 2. Materials and Methods

### 2.1. Mitochondrial Genomes

Complete mt genome sequences of salangid fishes were accessed from the Genetic Sequence Data Bank [97] (the National Center for Biotechnology Information; https://www.ncbi.nlm.nih.gov/; GenBank Flat File Release 244.0; see Appendix A for accession numbers). The outgroup species, including representatives of the genera *Plecoglossus* and *Retropinna*, were selected based on the previous molecular evidence of their close relationship to the family Salangidae [17,19,31] and the screening of nucleotide sequences available in GenBank.

### 2.2. DNA Sequence Analysis

The nucleotide sequences were aligned using the MUSCLE [98] and MAFFT v. 7 [99] software. The DnaSP v. 6 [100], PROSEQ v. 2.9 [101], and MEGA v. 7 [102] programs were used for intra- and interspecific analysis of polymorphism and divergence; MEGA v. 7 [102] was also used for basic phylogenetic analyses. Phylogenetic reconstructions were inferred from the analysis of the complete mt genomes by the maximum-likelihood methods available in IQ-TREE v. 2 [103,104,105]. The TIM2+F+I+G4 model showed the lowest Akaike Information Criterion (AIC; [106]) value (189,144.4265) and the Bayesian information criterion (BIC; [107]) score (189,529.5437); this model was selected for further phylogenetic reconstructions. The ultrafast maximum likelihood bootstrap analysis [108] consisted of 10,000 replicates. The alignments were analyzed for evidence of recombination using the pairwise homoplasy index (PHI) test [109] and various recombination detection methods provided in the RDP4 software [110,111,112,113,114,115,116,117,118]. The sliding window method (see, e.g., [119]) was used to examine the spatial distribution of polymorphism and divergence across the mt genomes studied. In this method, a window of specified length moves over the nucleotide sequence with a fixed size of step and the corresponding estimates are computed over the data in the window. The obtained estimates are assigned to the nucleotide position at the midpoint of the window. The output of the sliding window analysis can be presented graphically; the values of variation are plotted against the nucleotide position.

## 3. Results

### 3.1. Nucleotide Diversity and Divergence

A phylogram of the *P. chinensis* mt genomes along with those of other salangid fishes is displayed in Figure 1. The tree shows the *P. chinensis* specimens forming a single clade with two significantly different (100% bootstrap support) groups of sequences including (1) MW291629, KP306787, and HM106494, and (2) MH330683 and KJ499917 (Figure 1). The π value is quite high (0.0127 ± 0.0007). Despite the fact that it seems still below the range of divergence between species (e.g., [120]), it is, nevertheless, significantly higher than the level of intraspecific diversity detected for many other freshwater and marine fish species including, e.g., the taimen *Hucho taimen* (0.0010 ± 0.0002; [92]), the European whitefish *Coregonus lavaretus* (0.0025 ± 0.0003; [121]), the lake whitefish *C. clupeaformis* (0.0008 ± 0.0001; [121]), the houting *C. oxyrinchus* (0.0029 ± 0.0003; [121]), the Atlantic herring *Clupea harengus* (0.0061 ± 0.0008; [122]), the spotted wolffish *Anarhichas minor* (0.0005 ± 0.0001; [123]), and the northern wolffish *Anarhichas denticulatus* (0.0006 ± 0.0001; [123]).

### 3.2. Sliding Window Analysis

A sliding window analysis has revealed a non-uniform distribution of the nucleotide diversity values along the *P. chinensis* genomes characterized by four pronounced peaks centered at the (1) *COI*, (2) *ND4L*-*ND4* (hereinafter referred to as *ND4* for simplicity), and (3) *ND5* genes, and also at the (4) control region (CR) (Figure 2a). Comparisons including different pairs of the *P. chinensis* genomes have revealed distinct patterns with one to four strong peaks (Figure 2b–d,f–h), or no peaks at all (Figure 2e). The π values in the valley regions are low (varying from 0 to 0.0117), but follow much higher values in the high divergence (HD) regions: 0.0417 (first peak, *COI*; midpoint in the alignment 5631), 0.1217 (second peak, *ND4*; midpoint 11,557), 0.1100 (third peak, *ND5*; midpoint 13,096), and 0.1083 (fourth peak, CR; midpoint 15,763). The divergence values detected at the HD regions are significantly higher than the values usually observed on the intraspecific level (see above) and approach the intergeneric values, e.g., for scorpion fishes [120,124]. A more detailed interpretation for the distribution of divergent regions is considered below (Section 3.3).

The basic local alignment search tool (BLAST) procedure [125] has revealed a surprisingly high similarity (99–100%) between *P. chinensis* (KJ499917) and *N. tangkahkeii* (KP170510) for the HD regions 1 (*COI*), 2 (*ND4*), and 4 (CR); between *P. chinensis* (MH330683) and *N. tangkahkeii* (KP170510) for the HD region 2 (*ND4*); and between *P. chinensis* (HM106494) and *N. anderssoni* (HM106492) for the HD region 3 (*ND5*). Table 1 provides values of pairwise distances (*D*_xy_ ± SE) between *P. chinensis* and these closely related salangid species, and shows a number of outlier values obtained for the full mt genomes and the HD regions separately.

On the intraspecific level, the pairwise distances between the full mt genomes of *P. chinensis* vary within a fairly broad range (0.3–1.9%), which is less pronounced (0.1–0.4%) for the mt genomes with the HD regions deleted (Table 1, last column). When the HD regions are considered separately, the pairwise distances vary in a broader range: from zero to 11.6% (*ND4* and *ND5* regions), between 0.7–6.4% (*COI* region), and up to 12.9% (CR region) (Table 1). On the interspecific level, the pairwise distance fluctuations are even more pronounced. For instance, the full mt genome divergence between *P. chinensis* and *N. anderssoni* amounts to 5.5% with a range from 0.9% (*ND5* region) to 14.1% (CR region). For the *P. chinensis*–*N. tangkahkeii* pairwise comparisons, the average full mt divergence amounts to 8.5%. However, the *D*_xy_ value between these species is 85.0-fold lower (0.1%) for the *ND4* HD region. Moreover, for the *COI* and CR HD regions, there are no differences between *P. chinensis* (KJ499917) and *N. tangkahkeii* (KP170510) (Table 1). The pairwise distances inferred from the mt genomes with the HD regions deleted (Table 1, last column) show much lower fluctuations, ranging from 4.4 to 4.5% and from 8.6 to 8.7% for the *P. chinensis*–*N. anderssoni* and *P. chinensis*–*N. tangkahkeii* comparisons, respectively.

Thus, the HD regions of the *P. chinensis* genomes demonstrate an unexpectedly high similarity (*ND4* and *ND5*) or even full identity (*COI* and CR) to *N. tangkahkeii* and *N. anderssoni*, and can be explained by a recombination of mtDNA (Section 3.3). The four HD regions (*COI*, *ND4*, *ND5*, and CR) show sharply discordant phylogenetic signals between *P. chinensis* and the *Neosalanx* species. As a consequence, the position of *P. chinensis* is sharply different, depending on the fragments used for tree reconstruction (Figure 3). The trees inferred from the HD regions separately show *P. chinensis* as identical (or very similar) to *N. tangkahkeii* or to *N. anderssoni*, respectively (Figure 3a–d). In the tree without the HD regions (Figure 3e), the *P. chinensis* mt genomes (KJ499917, MH330683, HM106494, KP306787, and MW291629) are within a single cluster showing a value of π (0.0033 ± 0.0003) similar to that of the mt genomes of other fishes (Section 2.1). The other genera included in this analysis (*Salangichthys*, *Plecoglossus*, and *Retropinna*) do not show any visible discordance in the level of divergence between the HD regions and the rest of the mtDNA (Figure 3a–e). The presence or absence of the *N. tangkahkeii* and *N. anderssoni* fragments in the *P. chinensis* mt genomes erroneously increases or decreases the pairwise distances depending on their combination in particular comparison (Table 1). The same interpretation is suitable to explain the fluctuations in pairwise distances (Table 1).

### 3.3. Recombination

Based on the above-presented results, we suggest that the HD regions (Figure 2) and the respective phylogenetic inconsistencies (Figure 3) can be explained by recombination of the *P. chinensis* mt genome. Indeed, the PHI-test [109] has revealed very good evidence of recombination (*p* < 0.00001) in the alignment including the *P. chinensis*, *N. tangkahkeii*, and *N. anderssoni* mt genome sequences. We, therefore, have analyzed the mtDNA alignments for evidence of recombination using various recombination detection methods implemented in the RDP4 program ([110]; Table 2). Six methods have detected four recombination events in the *P. chinensis* KJ499917, MH330683, and HM106494 mt genomes with high statistical support (Table 2, Figure 4). The breakpoint positions (Table 2) match closely the coordinates of the HD regions (see Figure 2). The recombinant fragments are not fixed, and the recombinant mt genomes are mosaic, containing different numbers of recombinant events: three for KJ499917, one for MH330683, and one for HM106494 (Figure 4). No recombination events were detected for the *P. chinensis* KP306787 and MW291629 mt genomes.

The sequences involved in the recombination and the breakpoints were identified using the RDP4 suite [110] which incorporates the algorithms RDP (R), GENECONV (G), BOOTSCAN (B), MAXCHI (M), CHIMAERA (C), SISCAN (S), and 3SEQ (Q) (for references, see the Material and Methods section). For each putative recombination breakpoint, a Bonferroni correction *P*-value was calculated. For each recombination event involving the *COI*, *ND4*, and *ND5* gene regions, and also CR, the analysis was based on the alignment including three full mitochondrial genomes, as indicated in the column “Recombination event” (the length of the recombinant fragment is in parentheses). For the *ND4* gene region, two separate analyses were carried out, including the *Protosalanx chinensis* genomes KJ499917 (Recombinant 1) and MH330683 (Recombinant 2) as recombinant sequences with the same major and minor parents; conservative *P*-values are presented. The recombination events are arranged in the order as they occurred along the alignment. The breakpoint positions match closely the coordinates of the HD regions (see Figure 2). The *COI* recombinant fragment (*P. chinensis*, KJ499917) starts 73 bp downstream of the *COI* start codon and ends 1338 bp upstream of the *COI* stop codon. The *ND4* recombinant fragment (*P. chinensis*, KJ499917 and MH330683) starts 78 bp downstream of the *ND4L* start codon and covers the full *ND4* gene, tRNA-His, and 47 bp of the tRNA-Ser. The *ND5* recombinant fragment (*P. chinensis*, HM106494) starts 976 bp downstream of the *ND5* start codon and ends 423 bp upstream of the *ND5* stop codon. The CR recombinant fragment (*P. chinensis*; KJ499917) starts 6 bp downstream of the beginning of the tRNA-Pro, covers the rest part of the tRNA-Pro and includes 295 bp of the CR.

For the *COI* and CR recombinant fragments, there is 100% identity between the recombinant (*P. chinensis*, KJ499917) and the minor (*N. tangkahkeii*, KP170510) parent sequences (Figure 5a,e). However, for the *ND4* recombinant fragment, there are two mismatches at sites 10,174 and 10,185 (highlighted in bold in the alignment; Figure 5b,c) between the recombinant (*P. chinensis*, KJ499917, MH330683) and the minor (*N. tangkahkeii*, KP170510) parent sequences. Similarly, for the *ND5* recombinant fragment there are four mismatches at sites 12,954, 12,991, 12,997, and 13,030 between the recombinant (*P. chinensis*, HM106494) and the minor parent (*N. anderssoni*, HM106492) sequences (Figure 5d). See also Appendix A for the alignment of the variable sites in the recombinant and parent mt genome sequences. 

Taking into account the patterns of recombination revealed by the RDP4 program (Table 2, Figure 4), we can now confidently interpret the distribution of the HD regions in pairwise comparisons of the *P. chinensis* genomes obtained by the sliding window analysis (Figure 2) in the following way. The plot (a), inferred from the total data for the five *P. chinensis* mt genomes, shows four HD regions corresponding to the four recombination events detected for the three *P. chinensis* mt genomes: three for KJ499917 (*COI*, *ND4*, and CR), one for MH330683 (*ND4*), and one for HM106494 (*ND5*). The plots (b,c,h) showing one, two, and three HD regions, respectively, reflect the *COI*, *ND4*, and CR recombination events between the *P. chinensis* and *N. tangkahkeii* mt genomes. The plot (f) with a single HD region reflects the *ND5* recombination event involving *P. chinensis* and *N. anderssoni*. The plots (d,g) with four and two HD regions, respectively, reflect the *COI*, *ND4*, *ND5*, and CR recombination events involving *P. chinensis* and both *N. tangkahkeii* and *N. anderssoni*. The sliding window plot (e) shows the variability distribution for the two sequences (KP306787 and MW291629) lacking the recombinant fragments and, correspondingly, lacking any HD regions.

## 4. Discussion

Recombination of the mt genome can reflect interspecific hybridization; however, as an alternative hypothesis, it can be an artifact produced by polymerase chain reaction (PCR) or other errors. Below, we consider the most relevant alternative hypothesis which can explain the results observed in the present work.

Species identification of salangid fishes remains a serious challenge (Section 1.1). Consequently, one can argue that the GenBank data for *P. chinensis* and the close relatives, which we used for the present analyses, are wrong. Indeed, there are some relatively rare examples of incorrect taxonomic identification of fishes detected in GenBank [126,127,128]. However, all five *P. chinensis* mt genomes form a single cluster; the sequence divergence between them (except recombinant fragments) is low, 0.3% (Section 2.2; Figure 1 and Figure 4e), which closely matches the values of intraspecific diversity reported for many freshwater and marine fishes (Section 2.1). Furthermore, all the *P. chinensis* recombinant (KJ499917, MH330683, and HM106494) and non-recombinant (KP306787 and MW291629) mt genomes show close affinity to other salangid fishes, both in our analysis (Figure 1) and other publications [31,129]. Thus, the available data suggest that taxonomical misidentification is not responsible for the evidence of recombination obtained in our work.

Even if the taxonomic identification of *P. chinensis* is not doubted, there is still a risk of artificial recombination generated by PCR errors if more than one template were present in the PCR. In this case, the polymerase may jump from one template to another during the PCR, thus, producing artifactual recombinants [130]. This effect could be suggested for the *ND5* recombination involving the *P. chinensis* (HM106494) and *N. anderssoni* (HM106492) mt genomes, both investigated by Li et al. [31]. The multi-template PCR containing the *P. chinensis* DNA and *N. anderssoni* contamination DNA could generate PCR artifacts. However, if this recombination had occurred through PCR jumping, the exchanged regions found in two different sequences should have been exact copies of one another. An examination of the recombinant *P. chinensis* (HM106494) sequence has revealed no exact matches with the minor parent sequence (*N. anderssoni*; HM106492) for the *ND5* recombinant event. In particular, there are four mutations distinguishing the recombinant and minor parent sequences (Figure 5d), which makes the “jumping PCR” hypothesis highly unlikely. A similar pattern has been revealed for the *ND4* recombinant event (Figure 5b) with two mismatches along the recombinant sequence distinguishing the recombinant (*P. chinensis*; KJ499917, MH330683) and minor parent (*N. tangkahkeii*; KP170510) sequences. Thus, the imperfect homologies between the recombinants and minor parents for the *ND4* and *ND5* regions show that these recombinants are real and probably occurred some time ago, due to a historical hybridization between the icefishes.

The *COI* and CR recombinant sequences are identical to the minor parent sequences and could be explained by PCR jumping. Both recombinants are detected in the *P. chinensis* mt genome KJ499917 obtained by Lu et al. [30] (published online, 4 April 2014) from the Heilongjiang River Fisheries Research Institute, Chinese Academy of Fishery Sciences, Harbin, China. The fish sample was collected from Lake Xingkai (Khanka). However, the *N. tangkahkeii* mt genome KP170510, which represents a minor parent mt genome for the *COI* and CR recombination events, was sequenced by Zhong et al. [32] (published online 19 February 2015) from the Freshwater Fisheries Research Institute of Jiangsu Province, Nanjing, China. The fish sample was collected from Lake Taihu, Jiangsu Province, China. Thus, the *COI* and CR recombinants cannot be results of DNA contaminations of the *P. chinensis* PCR amplifications by the *N. tangkahkeii* DNA because these mt genomes were sequenced by different researchers at a different time and in different laboratories.

Another argument against PCR jumping is exemplified by two *P. chinensis* mt genome sequences, KJ400017 [30] and MH330683 [129], which carry out the identical recombination event including the *ND4* region (Figure 4a,b and Figure 5b,c). The recombination involves precisely the same region in both mt genomes (Table 2) which, however, were detected by different authors at different laboratories and in different years. The *P. chinensis* recombinant mt genome sequence KJ400017 was obtained by Lu et al. [30] (see the author’s affiliation and the fish sampling point above), but the *P. chinensis* recombinant sequence MH330683 was obtained by Liu et al. [129] from the Freshwater Fisheries Research Center, Chinese Academy of Fishery Sciences, Wuxi, China. The fish sample was collected from Lake Taihu, Wuxi City, China. It is difficult to imagine how this identical recombination could be obtained simultaneously and erroneously by different authors in different places and at different times.

It might also be suggested that the evidence of recombination obtained in the present study is caused by a problem with the tests of recombination due to, e.g., misalignments, which are a common cause of false positive recombination signals [110]. This explanation could be particularly relevant for the recombination detected within CR, which is the most variable part of the mt genome frequently characterized by multiple insertions, deletions, and the complex arrangement of tandem repeats [131]. However, the recombinant (*P. chinensis*, KJ499917) and minor parent (*N. tangkahkeii*, KP170510) sequences are 100% identical in the CR recombinant fragment, which provides fully unambiguous alignment and implies that the CR recombination that we have detected is unlikely to be the product of misalignment.

The above arguments strongly suggest that the mt genome recombinants detected in our work are not experimental errors or artifacts and, consequently, their formation requires assuming an interspecific hybridization between *P. chinensis* and the other two related salangid fishes, *N. tangkahkeii* and *N. anderssoni*. The necessary prerequisites for the mt genome recombination to occur, including paternal leakage and heteroplasmy, are not investigated for the salangids, but are well documented for interspecific hybrids of fishes (e.g., [94,132]) and many other organisms (reviewed by Ladoukakis and Zouros [84] and Parakatselaki et al. [85]). Mitochondrial fusion has been demonstrated in diverse organisms, from yeast to mammals including humans (review by Westermann [133]). Experiments testing the complementation between different mtDNA molecules after fusion have revealed the presence of an extensive and continuous exchange of genetic contents between mitochondria in mammalian cells [134]. Experimental data also show that mitochondria possess all the necessary molecular mechanisms for mtDNA recombination (review by Chen [135]).

The present data complement a number of examples of the mtDNA recombination reported previously for plants, fungi, protists, and animals (review in [82,84,95,135,136,137,138,139,140,141,142,143]). Estimation of hybridization events based on recombination can be considered a conservative one because, at least in animals, an overwhelming amount of mtDNA is maternally transmitted, and the paternal leakage is restricted to a very low amount [84]. The *P. chinensis* mtDNA recombination was detected in three out of the five mt genomes studied, thus, suggesting relatively frequent interspecific hybridization in salangid fishes, as it was detected, e.g., in hybrid swarms of other invasive species ([73]; see below).

Different species of salangid fishes frequently occur in sympatry [19,48,144]. Nevertheless, the probability of interbreeding between salangids is apparently low or negligible due to, e.g., different microhabitat preferences (such as salinity tolerances) or a shift of spawning season [9,48]. Indeed, Zhang et al. [19] reported a relatively high genetic divergence (based on the *CytB* gene) between salangid species and suggested that these fishes have “evolved powerful sorting mechanisms to maintain interspecific isolation” ([19], page 338). However, temporal or spatial reproductive isolation that is likely to occur in native populations (due to, e.g., habitat-based segregation at spawning grounds; [145]) can be lost in the non-native ranges, resulting in an extensive hybridization between invasive species in the absence of environmental cues present in their native range [73,78] (review by Scribner et al. [146]). Due to anthropogenic habitat disturbance, it may be that reinforcement does not produce enough efficient mechanisms to provide reproductive isolation for divergent species that naturally occur in sympatry [147].

Among salangids, *P. chinensis* and *N. tangkahkeii* exhibit the widest distribution and the highest ecological plasticity (review by Kang et al. [48]). After being introduced into a new habitat outside the natural distribution range, these species demonstrate flexible life-history traits, especially in their growth and reproductive modes [48,55], which are frequently observed in invasive species (e.g., [148,149,150,151]). The high reproductive plasticity and the lack of native environmental cues that would provide reproductive isolation may have contributed to the hybridization between *P. chinensis* and the *Neosalanx* species, as has been shown for other invasive fishes invading non-native ranges [70,71,72,73,74,75,77,78].

Interspecific hybridization between invasive species can be exemplified by one exceptionally problematic case: the introduction of silver carp (*Hypophthalmichthys molitrix*) and bighead carp (*H. nobilis*) to the complex system of braided watercourses in the Mississippi River Basin, United States [71,72,73]. These two species currently remain genetically isolated and do not crossbreed within their native ranges in China, but their introduction to a new habitat initiated an unprecedented frequency of interspecific hybridization leading to a hybrid swarm with early-generation hybrids driving the range expansion and contributing to their invasion success, which caused multiple ecological, evolutionary, and economic problems [73,77,78]. The introduction of invasive salangids into the new habitats of Lake Khanka and the Amur River Basin, located at the border between China and Russia [13,58,59,60,61,62,63], might have a similar hybridization potential, as is evidenced by the data obtained in the present work. For instance, three recombinant fragments were detected in the mt genome KJ400017 (Figure 4a) of *P. chinensis* sampled from Lake Khanka [30].

The genetic data obtained in the present study show clear signals of recombination in the mt genomes of *P. chinensis* containing four mtDNA fragments from two related salangid species, *N. tangkahkeii* and *N. anderssoni*. The recombinant fragments are not fixed, and the different mt genomes of *P. chinensis* are mosaic, containing a different number of recombinant events. These observations, along with the full identity of the *COI* and CR recombinant fragments between the donor and the recipient sequences, indicate the contemporary interspecific hybridization between *P. chinensis* and *N. tangkahkeii*, which can be at least partly explained by human-mediated activities involving the transplantation and introduction of the icefishes. Among the protein-coding mt genes, *COI* (along with *CytB*) exhibits the highest evolutionary rates, which are two–five-fold higher for the noncoding CR region [152]. These regions (*COI* and CR) in case of historical hybridization are expected to rapidly gain new mutations distinguishing them from the parental sequences. The *ND4* and *ND5* recombination events show two and four mismatches, respectively, between the recombinant and parent sequences (Figure 5b,d), which may indicate a historical hybridization between *P. chinensis* and the two related salangid fishes, *N. tangkahkeii* and *N. anderssoni*. Further evidence from biparentally inherited nuclear DNA is required to critically evaluate the revealed patterns and, in particular, to determine the fraction of the extant *P. chinensis* genome that was affected by the gene flow from the *Neosalanx* species, and the time when it happened.

The four recombinant fragments, including the *COI*, *ND4*, and *ND5* gene regions, and also the CR, are specific for distinguishing hybrids between *P. chinensis* and the two *Neosalanx* species. Two of them, *COI* and CR, that show full identity between *P. chinensis* and *N. tangkahkeii*, can be used as diagnostic markers to monitor the salangids’ human-mediated hybridization dynamics and their invasion fronts. Extensive spatial and temporal genetic sampling from both native and invasive ranges, including the Amur River Basin and Lake Khanka, should be carried out to determine the full extent of the current range of pure salangid species and their hybrids, which will help to elaborate the most appropriate aquaculture regimes to minimize the negative effects of the species introduction and transplantation on the local biodiversity and ecosystem resilience.

## Figures and Tables

**Figure 1 life-12-00661-f001:**
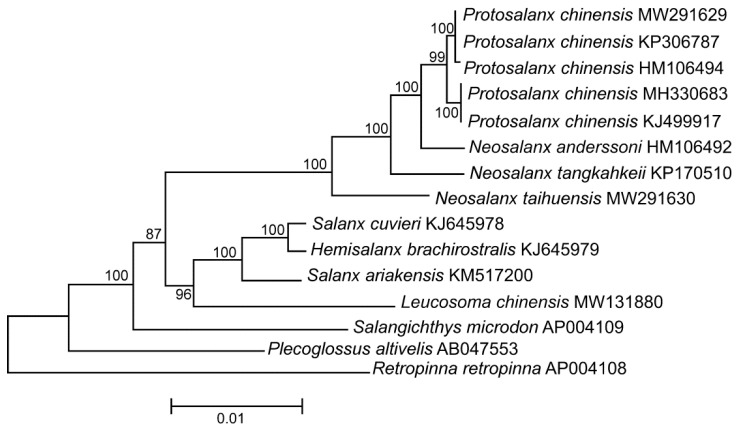
Maximum likelihood tree for the clearhead icefish *Protosalanx chinensis* and other members of the family Salangidae inferred from the complete mitochondrial (mt) genomes. The numerals at the nodes are bootstrap percent probability values based on 10,000 replications (values below 75% are omitted). *Plecoglossus altivelis* (Plecoglossidae) and *Retropiunna retropinna* (Retropinnidae) are used as outgroup. The species name *P. hyalocranius* was used for the originally published mt genome KJ499917 [30]. However, *P. hyalocranius* is a synonym of *P. chinensis* [19]. Consequently, we use the species name *P. chinensis* for the mt genome KJ499917 to avoid any confusion. The *P. chinensis* KP306787 genome is indicated as “Unverified” in GenBank. However, this mt genome shows very close affinity to all other *P. chinensis* genomes; the p-distance between this mt genome and the other *P. chinensis* mt genomes is 0.0111 ± 0.0007, which is even lower than the average p-distance for the rest of the *P. chinensis* mt genomes (0.0126 ± 0.0007). Consequently, we included the KP306787 mt genome in further analysis.

**Figure 2 life-12-00661-f002:**
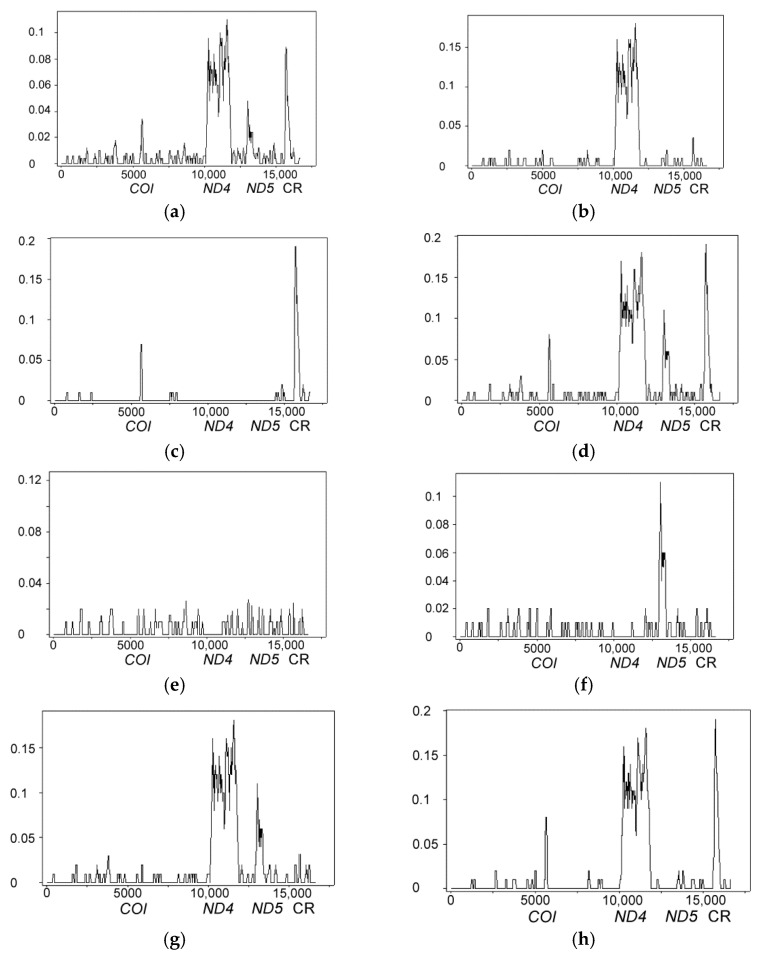
Sliding window plots of nucleotide variability along the complete mitochondrial genomes of *Protosalanx chinensis*. (**a**) Intraspecific polymorphism inferred from the five genomes studied; (**b**–**h**) Pairwise comparisons between the genomes are as follows: MW291629–MH330683 and KP306787–MH330683 (**b**); KJ499917–MH330683 (**c**); KJ499917–HM106494 (**d**); MW291629–KP306787 (**e**); MW291629–HM106494 (**f**); MH330683–HM106494 (**g**); MW291629–KJ499917 and KP306787–KJ499917 (**h**). Window sizes are 100 nucleotides with 25-nucleotide increments. Note the four significant peaks of divergence centered at the *COI*, *ND4*, and *ND5* genes, and also at CR.

**Figure 3 life-12-00661-f003:**
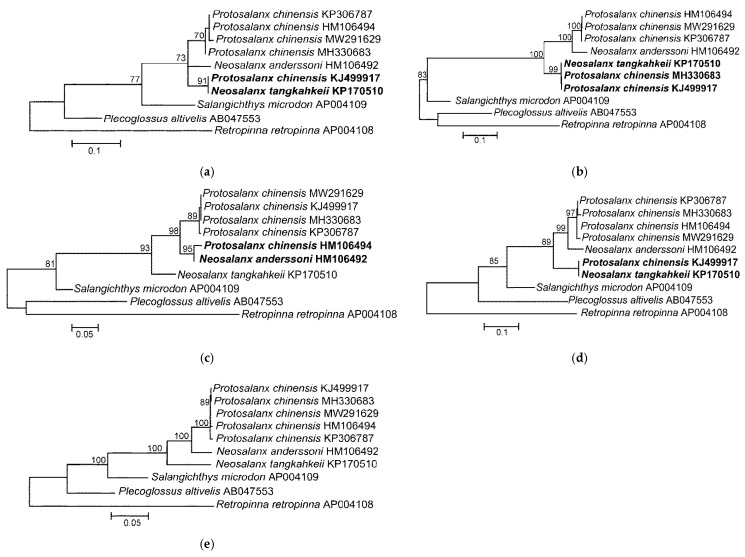
Maximum likelihood trees of the clearhead icefish *Protosalanx chinensis* and other members of the family Salangidae, as inferred from the analysis of the (**a**) *COI;* (**b**) *ND4;* (**c**) *ND5;* and (**d**) CR high divergence (HD) regions; and (**e**) the complete mitochondrial genomes with the HD regions excluded. The discordant phylogenetic signals between *P. chinensis* and the *Neosalanx* species are highlighted in bold. For other notes, see Figure 1.

**Figure 4 life-12-00661-f004:**
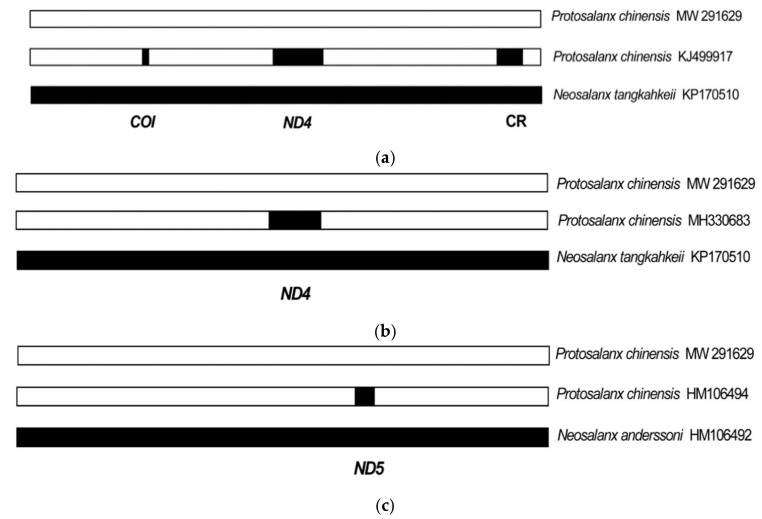
Schematic representation of the recombination events in the mitochondrial genomes of *Protosalanx chinensis* (**a**) KJ499917; (**b**) MH330683; and (**c**) HM106494. The major parental sequence (in white) is from *P. chinensis* (MW291629); the minor parental sequences (in black) are from *N. tangkahkeii* KP170510 (the *COI*, *ND4*, and CR recombination events) and *N. anderssoni* HM106492 (the *ND5* recombination event). The recombinant fragments are indicated by black boxes. For other notes, see Table 2.

**Figure 5 life-12-00661-f005:**
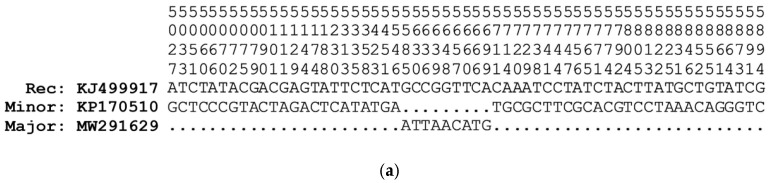
Alignment of variable nucleotide sites in the (**a**) *COI*; (**b**,**c**) *ND4*; (**d**) *ND5*; and (**e**) CR recombinant fragments and flanking regions for the salangid fishes *Protosalanx chinensis* (KJ499917, MW291629, and HM106494), *Neosalanx tangkahkeii* (KP170510), and *N. anderssoni* (HM106492). Dots indicate nucleotide identity with the recombinant sequence. Rec: recombinant sequence, Minor: minor parent sequence, Major: major parent sequence (see also Table 2 and Figure 4). The two mismatches in the *ND4* alignment and the four mismatches in the *ND5* alignment are highlighted in bold. The *ND4* recombination event is represented in two parts, illustrating the (**b**) beginning and (**c**) ending breakpoint regions, respectively. The alignments of variable sites in the *ND4* recombinant fragment are identical for the *P. chinensis* KJ499917 and MH330683 recombinant sequences. The polymorphic sites 10,391–11,558 of the *ND4* region alignment, containing no differences between the recombinant and the minor parent sequences, were deleted to fit onto the page.

**Table 1 life-12-00661-t001:** Pairwise distances (*D*_xy_ ± SE) between *Protosalanx chinensis* and closely related species inferred from different segments of mitochondrial genomes.

Species Pair	Full Genomes	COI	ND4	ND5	CR	HD Regions Excluded
*P. chinensis* MW291629—*P. chinensis* KJ499917	**0.0162 ± 0.0009**	**0.0638 ± 0.0196**	**0.1164 ± 0.0077**	0.0000 ± 0.0000	**0.1293 ± 0.0169**	0.0019 ± 0.0003
*P. chinensis* MW291629—*P. chinensis* MH330683	**0.0143 ± 0.0009**	**0.0142 ± 0.0096**	**0.1164 ± 0.0077**	0.0000 ± 0.0000	0.0201 ± 0.0071	0.0023 ± 0.0004
*P. chinensis* MW291629—*P. chinensis* HM106494	0.0047 ± 0.0005	0.0071 ± 0.0070	**0.0006 ± 0.0006**	**0.0682 ± 0.0116**	0.0057 ± 0.0040	0.0034 ± 0.0005
*P. chinensis* KJ499917—*P. chinensis* MH330683	0.0031 ± 0.0003	**0.0496 ± 0.0180**	0.0000 ± 0.0000	0.0000 ± 0.0000	**0.1264 ± 0.0169**	0.0011 ± 0.0002
*P. chinensis* KJ499917—*P. chinensis* HM106494	**0.0193 ± 0.0009**	**0.0567 ± 0.0192**	**0.1158 ± 0.0078**	**0.0682 ± 0.0116**	**0.1293 ± 0.0170**	0.0037 ± 0.0005
*P. chinensis* MH330683—*P. chinensis* HM106494	**0.0170 ± 0.0009**	0.0071 ± 0.0069	**0.1158 ± 0.0078**	**0.0682 ± 0.0116**	0.0144 ± 0.0059	0.0035 ± 0.0005
*P. chinensis* MW291629—*N. anderssoni* HM106492	0.0472 ± 0.0015	0.0496 ± 0.0186	0.0649 ± 0.0055	0.0588 ± 0.0108	0.0776 ± 0.0144	0.0439 ± 0.0018
*P. chinensis* KJ499917—*N. anderssoni* MH330683	0.0546 ± 0.0017	0.0638 ± 0.0196	0.1228 ± 0.0077	0.0588 ± 0.0108	**0.1408 ± 0.0174**	0.0446 ± 0.0018
*P. chineesis* MH330683—*N. anderssoni* HM106492	0.0540 ± 0.0016	0.0496 ± 0.0182	0.1228 ± 0.0077	0.0588 ± 0.0108	0.0833 ± 0.0145	0.0448 ± 0.0018
*P. chinensis* HM106494—*N. anderssoni* HM106492	0.0459 ± 0.0015	0.0426 ± 0.0173	0.0643 ± 0.0054	**0.0094 ± 0.0044**	0.0718 ± 0.0140	0.0442 ± 0.0017
*P. chinensis* MW291629—*N. tangkahkeii* KP170510	0.0893 ± 0.0018	0.0638 ± 0.0196	0.1152 ± 0.0077	0.1012 ± 0.0136	0.1293 ± 0.0169	0.0859 ± 0.0024
*P. chinensis* KJ499917—*N. tangkahkeii* KP170510	0.0753 ± 0.0018	**0.0000 ± 0.0000**	**0.0012 ± 0.0007**	0.1012 ± 0.0136	**0.0000 ± 0.0000**	0.0863 ± 0.0023
*P. chinensis* MH330683—*N. tangkahkeii* KP170510	0.0776 ± 0.0017	0.0496 ± 0.0180	**0.0012 ± 0.0007**	0.1012 ± 0.0136	0.1264 ± 0.0169	0.0864 ± 0.0025
*P. chinensis* HM106494—*N. tangkahkeii* KP170510	0.0899 ± 0.0019	0.0567 ± 0.0192	0.1146 ± 0.0078	0.1035 ± 0.0143	0.1293 ± 0.0170	0.0866 ± 0.0023

HD regions excluded: full mt genomes with deleted high divergence (HD) fragments located at the *COI*, *ND4*, *ND5*, and CR regions. The erroneously high or low pairwise p-distances due to the presence or absence of the *N. tangkahkeii* and *N. anderssoni* fragments in the *P. chinensis* genomes are highlighted in bold (see the text for details. The *P. chinensis* KP306787 mt genome with the redundant data concerning the pairwise distances are not included in the table.

**Table 2 life-12-00661-t002:** A summary of the recombination events detected in the *Protosalanx chinensis* mitochondrial genome.

Recombination Event	Beginning Breakpoint	Ending Breakpoint	*p*-Value
*COI* (140 bp) Recombinant: *Protosalanx chinensis* (KJ499917) Major parent: *Protosalanx chinensis* (MW291629) Minor parent: *Neosalanx tangkahkeii* (KP170510)	5566	5705	1.730 × 10^−13^ (G) 1.380 × 10^−14^ (B) 1.977 × 10^−02^ (M) 1.954 × 10^−02^ (C) 1.611 × 10^−05^ (Q)
*ND4* (1710 bp) Recombinant 1: *Protosalanx chinensis* (KJ499917) Recombinant 2: *Protosalanx chinensis* (MH330683) Major parent: *Protosalanx chinensis* (MW291629) Minor parent: *Neosalanx tangkahkeii* (KP170510)	10,167	11,876	3.865 × 10^−146^ (R) 6.784 × 10^−143^ (G) 2.238 × 10^−110^ (B) 3.884 × 10^−36^ (M) 2.448 × 10^−36^ (C) 4.440 × 10^−16^ (Q)
*ND5* (425 bp) Recombinant: *Protosalanx chinensis* (HM106494) Major parent: *Protosalanx chinensis* (MW291629) Minor parent: *Neosalanx anderssoni* (HM106492)	12,946	13,372	1.491 × 10^−26^ (R) 8.703 × 10^−25^ (G) 1.310 × 10^−26^ (B) 4.644 × 10^−10^ (M) 1.099 × 10^−09^ (C) 2.220 × 10^−16^ (Q)
CR (359 bp excluding alignment gaps) Recombinant: *Protosalanx chinensis* (KJ499917) Major parent: *Protosalanx chinensis* (MW291629) Minor parent: *Neosalanx tangkahkeii* (KP170510)	15,606	16,193	3.719 × 10^−49^ (R) 8.960 × 10^−55^ (G) 3.878 × 10^−49^ (B) 3.155 × 10^−13^ (M) 8.322 × 10^−13^ (C) 3.330 × 10^−16^ (Q)

## Data Availability

Not applicable.

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
