# Peer review of "Recombinant Mitochondrial Genomes Reveal Recent Interspecific Hybridization between Invasive Salangid Fishes"

_life, 2022, doi:10.3390/life12050661_

Round 1
Reviewer 1 Report
The manuscript entitled “Recombinant mitochondrial genomes reveal recent interspecific hybridization between invasive salangid fishes” describes the patterns of 13 diversity and recombination in the complete mitochondrial genomes of Protosalanx chinensis and other 11 commercially invasive salangid fishes in 12 China using the 14 GenBank resources, and provides useful insights about understanding interspecific recombination of these fishes. This work is a data analysis, including a comprehensive analysis of recombinant fragments, which revealed represent diagnostic genetic markers for identification and distinguishing of hybrids. Author has reviewed an intensive amount of literature and deduced many valuable results, with a full discussion section. Therefore, this communication is suitable to be published in Journal of life.
Author Response
Response to Reviewer 1 Comments
Thank you very much for the positive consideration of our manuscript.
Reviewer 2 Report
This is an impressive study, showing the recombination of mt genomes in Osteichthyes. However, some difficulties should be addressed before acceptance. 1) Explain the details of the sliding-window analysis in the materials and methods section. 2) Sequences look to be too different to identify orthologues between species (Fig. 5). Recheck the alignment of the sequences.
Author Response
Response to Reviewer 2 Comments
Thank you very much for giving us an opportunity to revise our manuscript. We have implemented your valuable suggestions. Please, find our point-by-point responses to your comments below.
Reviewer 2
“1) Explain the details of the sliding-window analysis in the materials and methods section.”
Response to Reviewer 2
We have added some details of the sliding-window analysis in the Materials and Methods section (see below).
“The sliding window method (see, e.g., [119]) was used to examine the spatial distribution of polymorphism and divergence across the mt genomes studied. In this method, a window of specified length moves over the nucleotide sequence with a fixed size of step and the corresponding estimates are computed over the data in the window. The obtained estimates are assigned to the nucleotide position at the midpoint of the window. The output of the sliding window analysis can be presented graphically; the values of variation are plotted against the nucleotide position.”
References
119. Hudson, R.R.; Kaplan, N.L. The coalescent process in models with selection and recombination. Genetics 1988, 120, 831–840.
Reviewer 2
“2) Sequences look to be too different to identify orthologues between species (Fig. 5). Recheck the alignment of the sequences.”
Response to Reviewer 2
Indeed, the sequences look to be too different to identify orthologues between species. However, when compared with the full mitochondrial genomes, these divergent fragments are very short (140 bp, 1,710 bp, 425 bp, and 359 bp for the COI, ND4, ND5, and CR fragments, respectively). We have rechecked the alignment strictly and added three new text files as Supplementary Materials (Text S1, Text S2, and Text S3) with the corresponding fully unambiguous alignments for the recombinant mitochondrial genomes.
This manuscript is a resubmission of an earlier submission. The following is a list of the peer review reports and author responses from that submission.
Round 1
Reviewer 1 Report
The conclusions drawn by the authors sounds quite interesting. However, interspecies recombination between two natural species has not been reported so far fro the maternally inherited mitochondrial genome. Introgression and hybridization between closely related species are common in nature, however, recombination of mitochondrial genome between species never reported. The author made the conclusion based on several sequences downloaded from the web source. We can see that there are always a minor part and a major parent for the so-called recombination. All the sequences were sequenced by traditional approach. PCR fragments of the genome and then combined them together. It is possible that the elements from the minor parent are artificial contaminations while the authors sequencing multiple species. That is to say, sequences from the minor parent were wrongly combined with the major ones. If there are common mitochondrial recombinations as reported by the author in this manuscript. The results could be favoured to be published in Science or Nature. So when the author could not verify the quality of the sequences used, it is highly dangerous to make a conclusion of interspecies recombination. I would definately encourage the author waiting for more evidence.
Reviewer 2 Report
Fishes of the family Salangidae comprise six genera and approximately 17 species. They are endemic to eastern Asia and distributed in coastal waters off Sakhalin, Vladivostok, Japan, the Korean Peninsula, mainland China, and northern Vietnam, as well as inland lakes and out-flowing rivers in China. Recently, biological invasion of two species of salangids has been paid specific attention. However, this manuscript has some serious questions.
- It is not fully reasonable to only use mitochondrial maker to text inter- specific hybridization ;
- Species identification of salangids appears to have some serious problems. This problem appears partly attributable to the morphological plasticity in this neotenic group, relating many misidentified species sequences in NCBI. Thus, all data from cited sequence in NCBI, has high possibility to have wrong result in this group.
- All salangids are annualism, but they show various habits and distribution patterns: most species are anadromous whereas some are restricted within certain regions of a few rivers. And they have quite different spawning season. Protosalanx chinensis spaws around January; Neosalanx tangkahkeii and anderssoni spawn around in April, but they have large genetic difference and prefer to various salinity. Sympatry is common in salangids, such as P. chinensis, Sx. ariakensis N. jordani, H. prognathus, and N. anderssoni in the estuary of the Yalujiang River; In most cases, sympatric salangids are not monophyletic in phlogeny, indicating that they are separated at an early stage in the evolution with consequent high genetic divergence, and their coexistence reflects secondary contact rather than sympatric speciation. On the other hand, if the sister species are sympatric (e.g. N. anderssoni and P. chinensis), they usually have different microhabitat preference (e.g. salinity tolerance) or a shift of spawning season to maintain reproductive isolation, having evolved powerful sorting mechanisms to maintain interspecific isolation. That is why I do not agree with the author. I think the author has not enough biological knowledge to discuss inter- specific hybridization;
- The author used the photo in Figure 1 with no permission of the original authors. This is a kind of academic behavior that is not allowed.